# Pyridine nucleotide redox potential in coronary smooth muscle couples myocardial blood flow to cardiac metabolism

Marc M. Dwenger[1], Sean M. Raph[1], Michelle L. Reyzer[2], M. Lisa Manier[2], Daniel W. Riggs[1], Zachary B. Wohl[1], Vahagn Ohanyan[3], Gregory Mack,[3], Thomas Pucci[3], Joseph B. Moore IV [1], Bradford G. Hill[1], William M. Chilian[3], Richard M. Caprioli[2], Aruni Bhatnagar[1] & Matthew A. Nystoriak [1✉]

Adequate oxygen delivery to the heart during stress is essential for sustaining cardiac function. Acute increases in myocardial oxygen demand evoke coronary vasodilation and enhance perfusion via functional upregulation of smooth muscle voltage-gated $K^+$ (Kv) channels. Because this response is controlled by Kv1 accessory subunits (i.e., Kvβ), which are NAD(P)(H)-dependent aldo-keto reductases, we tested the hypothesis that oxygen demand modifies arterial $[NAD(H)]_i$, and that resultant cytosolic pyridine nucleotide redox state influences Kv1 activity. High-resolution imaging mass spectrometry and live-cell imaging reveal cardiac workload-dependent increases in $NADH:NAD^+$ in intramyocardial arterial myocytes. Intracellular NAD(P)(H) redox ratios reflecting elevated oxygen demand potentiate native coronary Kv1 activity in a Kvβ2-dependent manner. Ablation of Kvβ2 catalysis suppresses redox-dependent increases in Kv1 activity, vasodilation, and the relationship between cardiac workload and myocardial blood flow. Collectively, this work suggests that the pyridine nucleotide sensitivity and enzymatic activity of Kvβ2 controls coronary vasoreactivity and myocardial blood flow during metabolic stress.

[1] Department of Medicine, Division of Environmental Medicine, Diabetes and Obesity Center, University of Louisville, Louisville, KY 40202, USA. [2] Department of Biochemistry, Vanderbilt University School of Medicine, Nashville, TN 37235, USA. [3] Department of Integrative Medical Sciences, Northeast Ohio Medical University, Rootstown, OH 44272, USA. ✉email: matthew.nystoriak@louisville.edu

Blood flow to the heart is tightly coupled with cardiac workload and oxygen consumption[1,2]. Acute elevations of heart rate and myocardial contractile force, such as those that occur during a bout of strenuous physical activity, are associated with a higher rate of cardiomyocyte oxidative ATP production to support ion transport and sarcomeric function.[3] Although oxygen extraction from the arterial blood supply is near maximal at rest, sustained metabolic activity of the heart during stress requires a near instantaneous reduction in coronary resistance to increase myocardial perfusion[4]. Despite the importance of maintaining adequate myocardial oxygen delivery in preventing myocardial ischemia, molecular mechanisms that underlie the physiologic regulation of coronary blood flow remain poorly understood.

Local metabolic control of coronary arterial tone occurs in part via complex intercellular signaling between active cardiomyocytes and intramyocardial vasculature[5]. In this process, multiple factors directly or indirectly modify the activity of ion channels in the sarcolemma of coronary smooth muscle cells (i.e., coronary arterial myocytes)[1,2,6]. In particular, activation of the redox-sensitive voltage-gated potassium channels (i.e., Kv1) in coronary arterial myocytes, upon increased myocardial oxygen demand, promotes rapid membrane hyperpolarization, reduced intracellular $[Ca^{2+}]$, and vasodilation[7–12]. The native Kv1 holo-channel structure consists of four transmembrane pore-forming subunits that interface with four intracellular Kvβ subunits[13,14]. We have previously found that selective modifications to the Kvβ complex composition impact vasoreactivity and myocardial blood flow regulation; in particular, the loss of Kvβ2, or conversely the overabundance of Kvβ1.1 in vascular Kv1 channels results in suppression of metabolic vasodilatory function, and thereby disrupts the relationship between myocardial blood flow and cardiac workload[10]. Nonetheless, the cellular responses to metabolic stress in the coronary circulation that ultimately promote Kvβ-dependent stimulation of Kv1 activity are unknown.

The Kvβ proteins are functionally active aldo-keto reductases (AKRs) that bind pyridine nucleotides (i.e., NAD(P)(H)) with high affinity. These proteins regulate Kv1 function in heterologous systems as well as native excitable cells[15–18]. Accordingly, we tested the hypothesis that changes in myocardial oxygen consumption and demand modulate the redox ratio of pyridine nucleotides in arterial smooth muscle and thereby influence the regulatory mode of Kvβ proteins and Kv activity. Using a combination of in vivo and ex vivo approaches, we show that the intracellular NADH:NAD$^+$ ratio in arterial smooth muscle is sensitive to changes in cardiac workload and that variable NAD(P)H:NAD(P)$^+$ ratios, reflective of enhanced myocardial metabolic demand, potentiate Kv1 activity in a Kvβ2-dependent manner. These findings suggest that a catalytically active Kvβ2 is essential for the metabolic control of myocardial blood flow, underscoring the importance of AKR enzymatic properties of Kvβ in cardiovascular physiology.

## Results

**Cardiac workload-dependent changes in vascular pyridine nucleotide redox potential**. To test whether acute changes in cardiac workload and metabolic demand impact the redox state of the myocardium and coronary arterial wall in vivo, we used high spatial resolution imaging mass spectrometry (IMS) to visualize and compare relative levels of lactate and pyruvate in hearts of mice subjected to short-term (i.e., ~5 min) elevation of cardiac workload (high workload) with those from low workload control mice (see Fig. 1A)[19]. The intracellular lactate:pyruvate ratio is a sensitive surrogate indicator of redox state of the pyridine nucleotide pair - NADH:NAD$^+$; both ratios are highly sensitive to oxygen levels in tissues and rise dramatically during ischemia[20]. Hence, we tested

whether transient changes of myocardial O$_2$ consumption caused by elevated workload modulate lactate:pyruvate in the heart and coronary vasculature. For this, we subjected mice to stress via administration of the β-adrenoceptor agonist dobutamine (10 mg/kg, i.p.) and hearts were frozen immediately (Fig. 1B). Branches of the left anterior descending coronary arteries were identified in H&E-stained sections as regions of interest for IMS measurements. Exemplary IMS images for lactate (m/z 89.024), pyruvate (m/z 87.009), and merged lactate:pyruvate, in hearts from low and high workload mice are shown in Fig. 1C. Comparison of signal intensities revealed significantly higher lactate:pyruvate in hearts from high workload mice relative to those in low workload mice both in the perivascular region and the coronary wall (Fig. 1C, D). These data suggest that an acute increase in cardiac workload elevates the lactate:pyruvate ratio in the intramyocardial vasculature, consistent with a rise in intracellular levels of NADH relative to NAD$^+$.

To determine directly whether the NADH:NAD$^+$ ratio in arterial smooth muscle cells is sensitive to local cardiomyocyte contractile function, we monitored cytosolic NADH:NAD$^+$ ratio using the genetically-encoded fluorescent biosensor peredox-mCherry (see Supplementary Fig. 1A)[21,22]. Consistent with previous reports, we observed a reduction in peredox-mCherry green:red fluorescence in the presence of decreasing external lactate:pyruvate in arterial myocytes (Supplementary Fig. 1). At baseline, the cytosolic NADH:NAD$^+$ ratio estimated by this technique was 0.0027 ± 0.0001, which is consistent with previous estimates made in vascular smooth muscle of porcine carotid strips[23], and indicated that most of the nucleotide is in its oxidized form (NAD$^+$). To examine whether the NADH:NAD$^+$ ratio in arterial myocytes is affected by the metabolic activity of cardiomyocytes, we seeded isolated arterial myocytes expressing peredox-mCherry on two-dimensional monolayers of induced pluripotent stem cell-derived cardiomyocytes (iPSC-CMs; Fig. 2A). After incorporation of peredox-mCherry-positive arterial myocytes into iPSC-CM monolayers (Fig. 2B), we monitored NADH:NAD$^+$ in arterial myocytes during step-wise increases in the frequency of electrical stimulation (1-3 Hz). Consistent with the redox modifications resulting from changes in cardiac workload seen in vivo (Fig. 1), we observed a significant increase in arterial myocyte cytosolic NADH:NAD$^+$ as the pacing frequency was increased (Fig. 2C, D). These frequency-dependent changes in NADH:NAD$^+$ were dependent on the presence of cardiomyocytes (frequency*group interaction, cardiac/arterial myocyte vs. arterial myocytes alone, p = 0.0007, Fig. 2C, D). Electrical pacing of co-cultures increased cytosolic NADH:NAD$^+$ in arterial myocytes to a similar degree as a reduction in chamber O$_2$ levels from 5 to 1% (Supplementary Fig. 2). Moreover, this elevation of arterial myocyte NADH:NAD$^+$ was prevented by the general redox cycling agent 4-hydroxy TEMPO (tempol; 1 mM; Supplementary Fig. 3), which scavenges reactive oxygen species (ROS), suggesting that this effect is at least partially dependent on ROS derived from cardiomyocytes[24]. To test the possibility that the changes in pyridine nucleotide ratios seen in vivo after β-adrenergic stimulation (Fig. 1) were due to direct agonist effects in arterial myocytes, we tested whether activation of β-adrenoceptors on arterial myocytes alters the cytosolic NADH:NAD$^+$ ratio. However, application of the synthetic catecholamine isoproterenol modestly reduced (e.g., 0.0018 ± 0.0010 in presence of 1 μM isoproterenol vs. 0.0022 ± 0.0001 at baseline), rather than enhanced, the NADH:NAD$^+$ ratio (Supplementary Fig. 4). Taken together, these data suggest that inotropic and chronotropic stimulation of cardiomyocytes elevates NADH:NAD$^+$ in adjacent arterial cells.

**Intracellular pyridine nucleotide redox states reflecting augmented O$_2$ demand increase coronary Kv1 activity**. Coronary arterial myocyte Kv1 channels are essential for proper metabolic

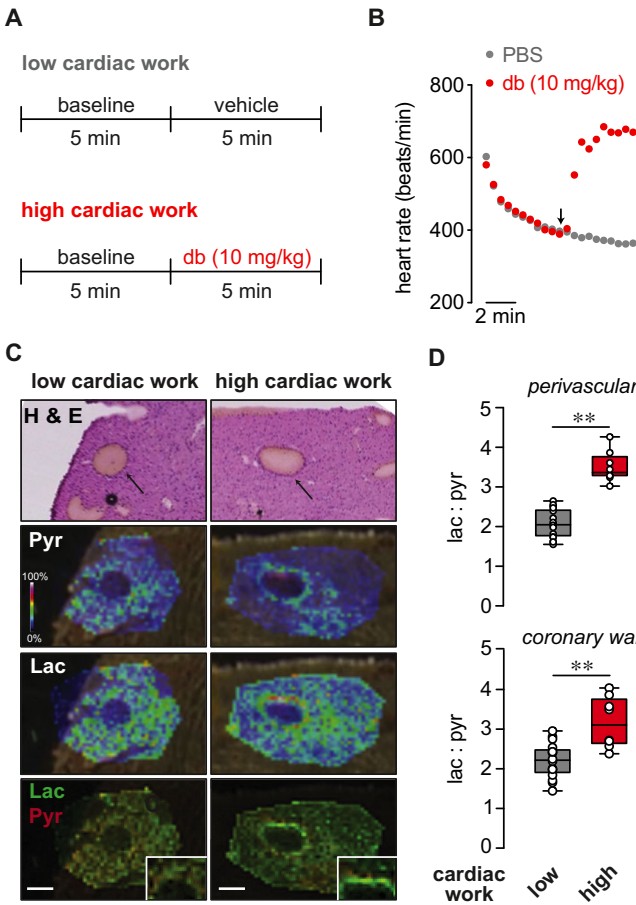

**Fig. 1 Increased myocardial workload promotes local elevation of lactate:pyruvate ratio in the coronary arterial wall. A** Schematic showing protocol for induction of acute cardiac stress prior to collection of tissues for MALDI-MS imaging. To induce high cardiac work, anesthetized mice with stable heart rates of 400–450 bpm were treated with dobutamine (db; 10 mg/kg, i.p.). Heart rate was continuously monitored, and hearts were rapidly cryopreserved in the thoracic cavity and excised after 2–3 min of stabilized heart rate responses to either db or PBS (vehicle). **B** Symbol plot showing exemplary heart rates recorded every 30 s before (i.e., during induction and stabilization of anesthesia) and after administration of either db (10 mg/kg; i.p.) or PBS. **C** Images of left ventricular intramyocardial coronary arteries in H & E-stained heart sections and corresponding intensity-coded MALDI-MS images showing lactate (Lac) and pyruvate (Pyr) relative to background signals in hearts from low and high cardiac work mice. $x–y$ resolution: ~20 μm. Insets show magnified region of interest at coronary arterial wall. Scale bars represent 200 μm. Experiment was repeated twice with similar results. **D** Box and whiskers plots (line: median, box: 25th to 75th percentile, whiskers: min and max) showing lactate:pyruvate ratios in perivascular myocardium (region of interest within ~250 μm from coronary wall; *top*) and coronary wall (*bottom*) in hearts of low and high cardiac work mice. (low cardiac work: $n = 16$ technical replicates, 4 mice, high cardiac work group: $n = 8$ technical replicates, 2 mice), perivascular, $**p < 0.0001$, coronary, $**p = 0.0003$ (unpaired two-sided $t$ test). Source data are provided as a Source Data file.

hyperemia[7,9,10,25,26]. Under native conditions, these channels associate with ancillary Kvβ proteins, which bind pyridine nucleotides with high affinity[15,27]. Based on results presented in Figs. 1 and 2, we tested whether pyridine nucleotide levels in smooth muscle that simulate altered myocardial $O_2$ demand could influence Kv1 activity. For this, using the conventional whole-cell configuration of the patch clamp technique, we recorded voltage-dependent outward $K^+$ currents after internal dialysis of cells with altered NAD(P)H:NAD(P)$^+$ ratios, referred to hereafter as either "oxidized" or "reduced" nucleotide compositions (see Supplementary Table 2 for concentrations and ratios). Under the applied patch conditions and nucleotide-free internal solution, the Kv1-selective inhibitor psora-4 (500 nM) inhibited ~60% of the total outward $K^+$ current recorded from isolated coronary arterial myocytes (Supplementary Fig. 5). As shown in Fig. 3A, the magnitude of $I_K$ recorded from coronary arterial myocytes perfused with reduced nucleotides was significantly higher than the current recorded from myocytes perfused with oxidized nucleotides (Fig. 3A, B—i.). No significant differences in $I_K$ density were observed between groups when the recordings were performed in the presence of 500 nM psora-4 (Fig. 3A, B—ii.). Representative psora-4-sensitive $I_K$ and summarized current densities in the presence of either oxidized or reduced nucleotide ratios are shown in Fig. 3A, B—iii. Reduced nucleotide conditions also promoted hyperpolarizing shifts in the voltage-sensitivity of activation and inactivation ($V_{0.5,act}$ and $V_{0.5,inact}$, respectively; Fig. 3C and Supplementary Table 3). These data support the notion that the intracellular redox ratio of pyridine nucleotides is a critical regulator of Kv1 activity, and that pyridine nucleotides at levels expected under conditions of high $O_2$ demand augment Kv1 current density.

In its native state, Kv1 associates with intracellular Kvβ, which is a key regulator of Kv1 gating. In the coronary vasculature, loss of the Kvβ2 subunit hampers coronary vasodilatory function and suppresses myocardial blood flow[10]. In arterial myocytes, loss of Kvβ2 does not significantly impact basal $I_K$ density, but results in modest shifts in $V_{0.5,act}$ and $V_{0.5,inact}$ (Supplementary Fig. 6). However, deletion of Kvβ2 completely abolished the pyridine nucleotide sensitivity of Kv current. In contrast to the effects observed in coronary arterial myocytes from wild-type mice, no differences in $I_K$ density were observed between oxidized and reduced nucleotide conditions in cells from Kvβ2$^{-/-}$ mice (Fig. 3D). In addition, the loss of Kvβ2 proteins markedly altered the response in voltage-sensitivity to reduced nucleotides; whereas robust hyperpolarizing shifts in $V_{0.5,act}$ and $V_{0.5,inact}$ were observed in cells from wild type mice (Fig. 3C), no depolarizing shifts were observed in $V_{0.5,act}$ and $V_{0.5,inact}$ in arterial myocytes from Kvβ2$^{-/-}$ mice (Fig. 3E and Supplementary Table 3). Collectively, these data indicate that in coronary arterial myocytes, Kv1 channels are sensitive to the redox state of pyridine nucleotides due to the presence of Kvβ2.

**Direct potentiation of native coronary Kv1 channel activity by NADH.** We next tested whether an elevation of NADH in the cytosolic compartment would affect the activity of native coronary Kv1 channels. To test this, we measured the open probability of single Kv channels using the inside-out patch configuration. Unitary current amplitudes recorded over a range of voltage in the presence of $K_{ATP}$ and $BK_{Ca}$ channel inhibitors (i.e., glibenclamide and iberiotoxin, respectively) showed similar conductance values between patches from freshly isolated coronary arterial myocytes and Cos-7 cells expressing Kv1.5 (Fig. 4A, B). Moreover, single channel events with amplitudes similar to that reported for Kv1 channels (+40 mV)[28] were abolished by application of 500 nM psora-4 in the bath solution (Fig. 4C, D), further supporting that channel activity observed under the specified conditions is mediated by Kv1 channels. At a holding potential of −40 mV, application of 1 mM NADH in the bath resulted in an immediate increase in the open probability of Kv1 channels ($nP_o$, Fig. 4E, F). In contrast, no change in $nP_o$ was observed in the presence of NAD$^+$, suggesting that nucleotide-induced changes in Kv1 activity are dependent on the redox state

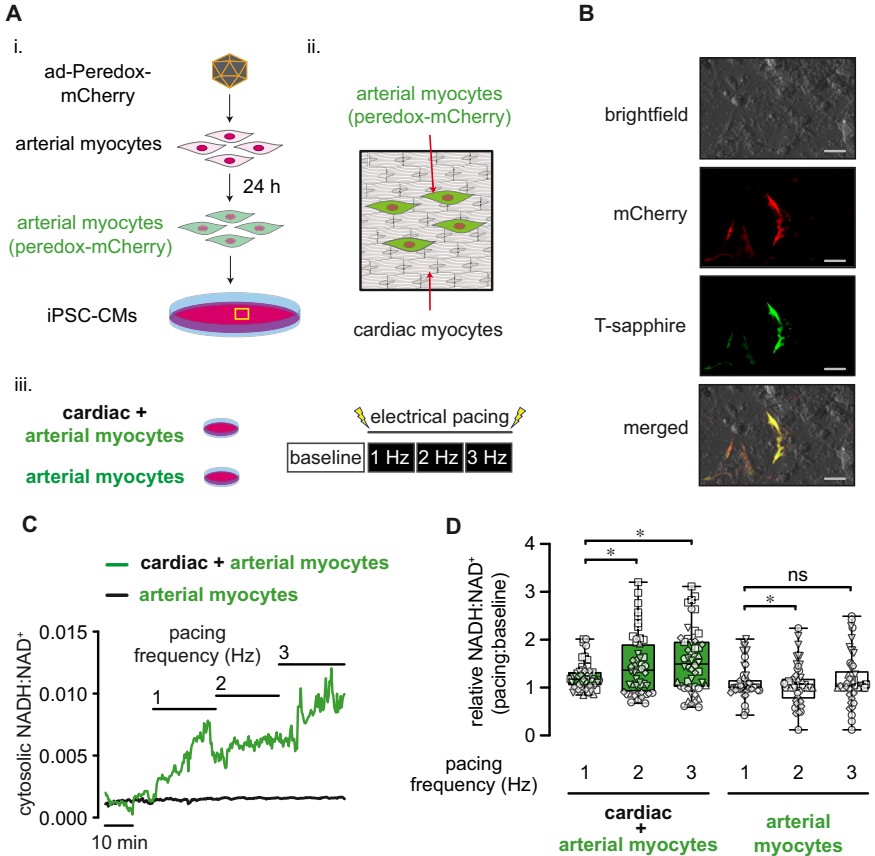

**Fig. 2 Arterial myocyte NADH:NAD⁺ is sensitive to changes in proximal cardiomyocyte beating frequency. A** Schematic illustration depicting the preparation of arterial myocyte/induced pluripotent stem cell-derived cardiomyocyte (iPSC-CM) co-cultures. Primary aortic vascular smooth muscle cells (passage 0–2) were treated with adenovirus to express the NADH:NAD⁺-sensitive fluorescent biosensor peredox-mCherry (i.). Arterial myocytes were then seeded and allowed to integrate (48 h) onto a two-dimensional monolayer of iPSC-CMs (ii.). Imaging was performed on either arterial/cardiac co-cultured myocytes or arterial myocytes alone during electrical stimulation (1–3 Hz; iii.) while monitoring peredox-mCherry green:red fluorescence. **B** Exemplary brightfield and fluorescence images showing red (mCherry) and green (T-sapphire) fluorescence of peredox-mCherry expressing arterial myocytes in the presence of non-fluorescent cardiac myocytes in an arterial + cardiac myocyte co-culture. Scale bars represent 100 μm. Experiment was repeated six times with similar results. **C** Exemplary time series of arterial myocyte NADH:NAD⁺ in either an arterial/cardiac myocyte co-culture (green trace) or arterial myocytes alone (– cardiac myocytes; black trace) at baseline (0 Hz) and during electrical stimulation (1–3 Hz). **D** Box and whiskers plots (line: median, box: 25th to 75th percentile, whiskers: min and max) summarizing fold-change in NADH:NAD⁺ in arterial myocytes in arterial/cardiac myocyte co-cultures or arterial myocytes alone. Arterial/cardiac myocytes, $n = 50$ cells from 5 independent experiments, 2 vs. 1 Hz, *$p = 0.0047$; 3 vs. 1 Hz, *$p = 0.0012$; Arterial myocytes only, $n = 52$ cells from 6 independent experiments, 2 vs. 1 Hz, *$p = 0.0029$; 3 vs. 1 Hz, ns: $p = 0.6574$ (Linear mixed models, log-transformed NADH:NAD⁺). Source data are provided as a Source Data file.

of pyridine nucleotides. Similar responses in channel activity in response to 1 mM NADH were observed in excised membrane patches from human coronary arterial myocytes (Fig. 4G). Note that using in situ proximity ligation to assess Kv1 subunit interactions as previously described, we found that freshly isolated human coronary arterial myocytes, like murine cells, express Kv1 pore-forming subunits that interact with Kvβ1.1 and Kvβ2 proteins (Supplementary Fig. 7)[10,18,27]. Whereas the NADH-induced increase in open probability was unaffected by the ablation of Kvβ1.1, this effect was abolished in coronary arterial myocyte membrane patches from mice lacking Kvβ2 (Fig. 4H, I). Together, these results indicate that NADH directly increases the activity of native coronary Kv1 channels and that the regulation of Kv1 by NADH is attributable to the Kvβ2 subunit.

**Redox control of Kv1 activity, vasodilation, and myocardial blood flow require intact enzymatic function of Kvβ2.** Considering that myocardial metabolism modifies coronary arterial myocyte pyridine nucleotide redox state and that this regulates

Kv1 activity via the Kvβ2 subunit, we next tested whether this mode of regulation involves Kvβ2 mediated catalysis. Previous work has shown that the Kvβ2 protein possesses weak aldehyde reductase activity that depends on a tyrosine residue at position 90 (Y90) for hydride transfer during the reductive catalytic cycle[16,29]. Therefore, to examine the role of Kvβ2-catalysis, we isolated arterial myocytes from mice in which the catalytic site tyrosine (Y90) of Kvβ2 is mutated to phenylalanine (Kvβ2^Y90F; Fig. 5A). The abundance of Kvβ2 protein was not significantly different between arteries from Kvβ2^Y90F mice compared with those from wild-type mice (Supplementary Fig. 8). In contrast to the NADH-induced increases in single Kv1 channel activity in coronary arterial myocytes from wild-type mice (Fig. 4E, F), no significant change in nP₀ was observed upon application of NADH in coronary arterial myocytes from Kvβ2^Y90F mice (Fig. 5B, C). These observations suggest that catalytic turnover is essential for the redox regulation of Kv1 currents by Kvβ2. Next, we examined whether catalytic turnover is also required for redox-dependent vasoreactivity and regulation of myocardial blood flow in vivo. We previously reported that ablation of

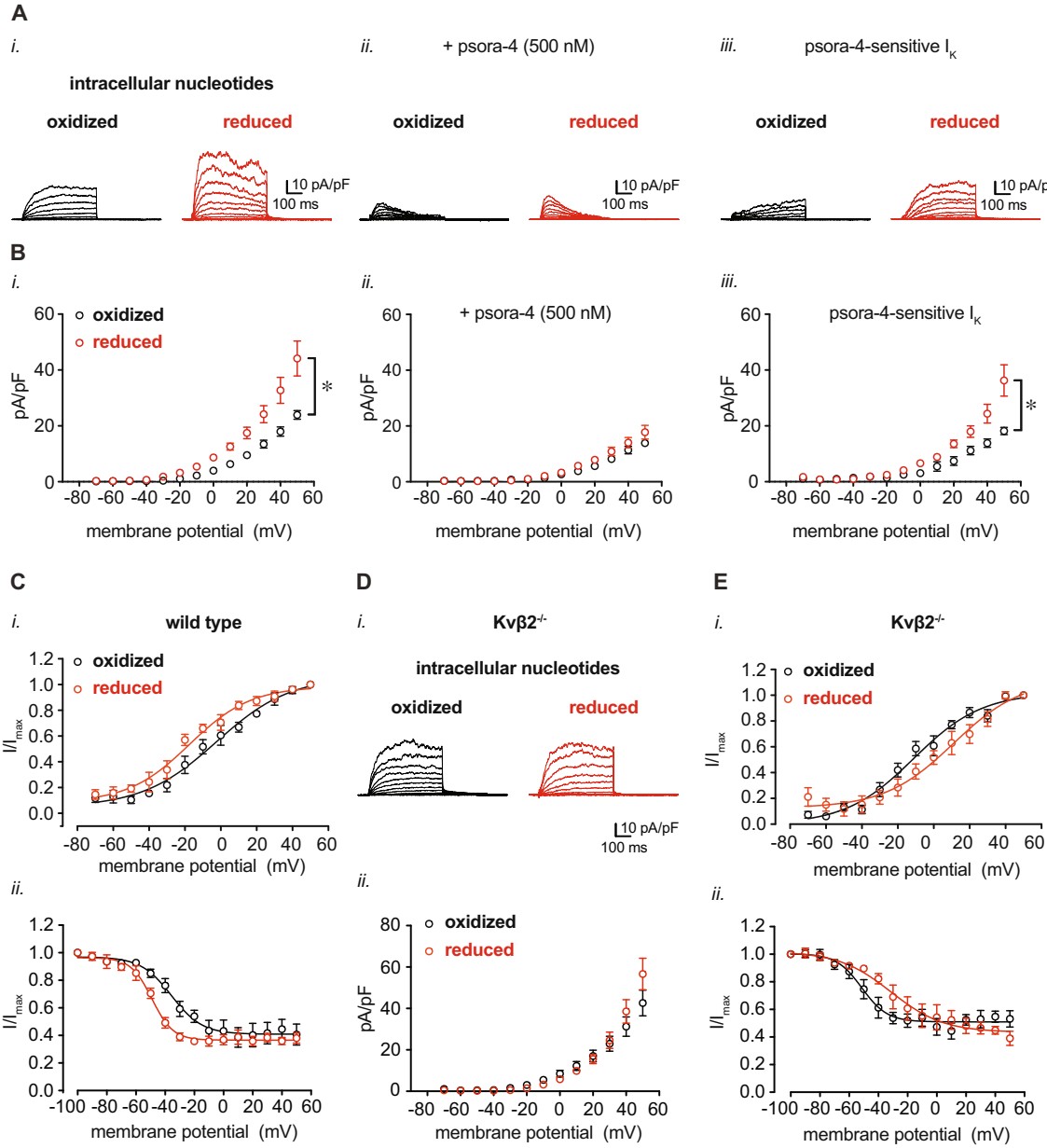

**Fig. 3 Modulation of coronary arterial myocyte $I_{Kv}$ upon changes in intracellular pyridine nucleotide redox potential.** Isolated coronary arterial myocytes were dialyzed with pyridine nucleotides at concentrations as indicated in Supplementary Table 2 for voltage-clamp recordings in the conventional whole-cell configuration. **A**, **B** Representative outward K+ currents (**A**) and $I_K$ densities (pA/pF; **B**) recorded in coronary arterial myocytes from wild type mice (129SvEv) in the presence of either oxidized or reduced pyridine nucleotide redox ratios. Recordings were performed in the absence (i.) and presence (ii.) of the Kv1 channel inhibitor psora-4 (500 nM). Representative psora-4-sensitive currents (i.e., total outward $I_K$ − psora-insensitive $I_K$) and summarized densities (mean values ± SEM) are shown in iii. panels. $n = 5$–6 cells, 4–5 mice for each. *$P < 0.001$, oxidized vs. reduced (mixed effects analysis). **C** Summarized $I/I_{max}$ (mean values ± SEM) from two-pulse activation voltage protocol (i.) and inactivation protocol (ii.) for coronary arterial myocytes from wild type mice in the presence of oxidized or reduced pyridine nucleotide ratios. Curves were fit with a Boltzmann function; $V_{0.5,act}$ and $V_{0.5,inact}$ are provided in Supplementary Table 3. $n = 5$–6 cells, 4–5 mice for each. **D** Representative total outward $I_K$ and summarized $I_K$ density (mean values ± SEM), as in A, recorded in coronary arterial myocytes from $Kv\beta2^{-/-}$ mice in the presence of either oxidized or reduced pyridine nucleotide ratios. $n = 8$–9 cells, 5 mice for each. (**E**) Plots showing summarized $I/I_{max}$ (mean values ± SEM) with Boltzmann fittings, as in (**D**), recorded from coronary arterial myocytes from $Kv\beta2^{-/-}$ mice. $V_{0.5,act}$ and $V_{0.5,inact}$ for each condition are provided in Supplementary Table 3. $n = 5$–7 cells, 4–5 mice for each. Source data are provided as a Source Data file.

Kvβ2 suppresses redox-dependent vasodilation induced by elevated external L-lactate[10]. In agreement with these observations, we found that elevation of L-lactate caused vasodilation in arteries isolated from wild type mice; however, this effect was abolished in arteries from $Kv\beta2^{Y90F}$ mice (Fig. 5D), indicating that Kv1-mediated vasodilation in response to changes in intracellular pyridine nucleotides depends entirely upon the catalytic activity

of Kvβ2. Moreover, vasodilation upon application of $H_2O_2$ (0.1–10 μM) was abolished in arteries from $Kv\beta2^{Y90F}$ mice (Fig. 5E), consistent with the role for Kvβ2 catalytic function in the physiologic vascular response to ROS production.

Next, to investigate whether Kv1-mediated regulation of myocardial blood flow depends similarly on catalytically active Kvβ2, we examined blood flow across a range of cardiac

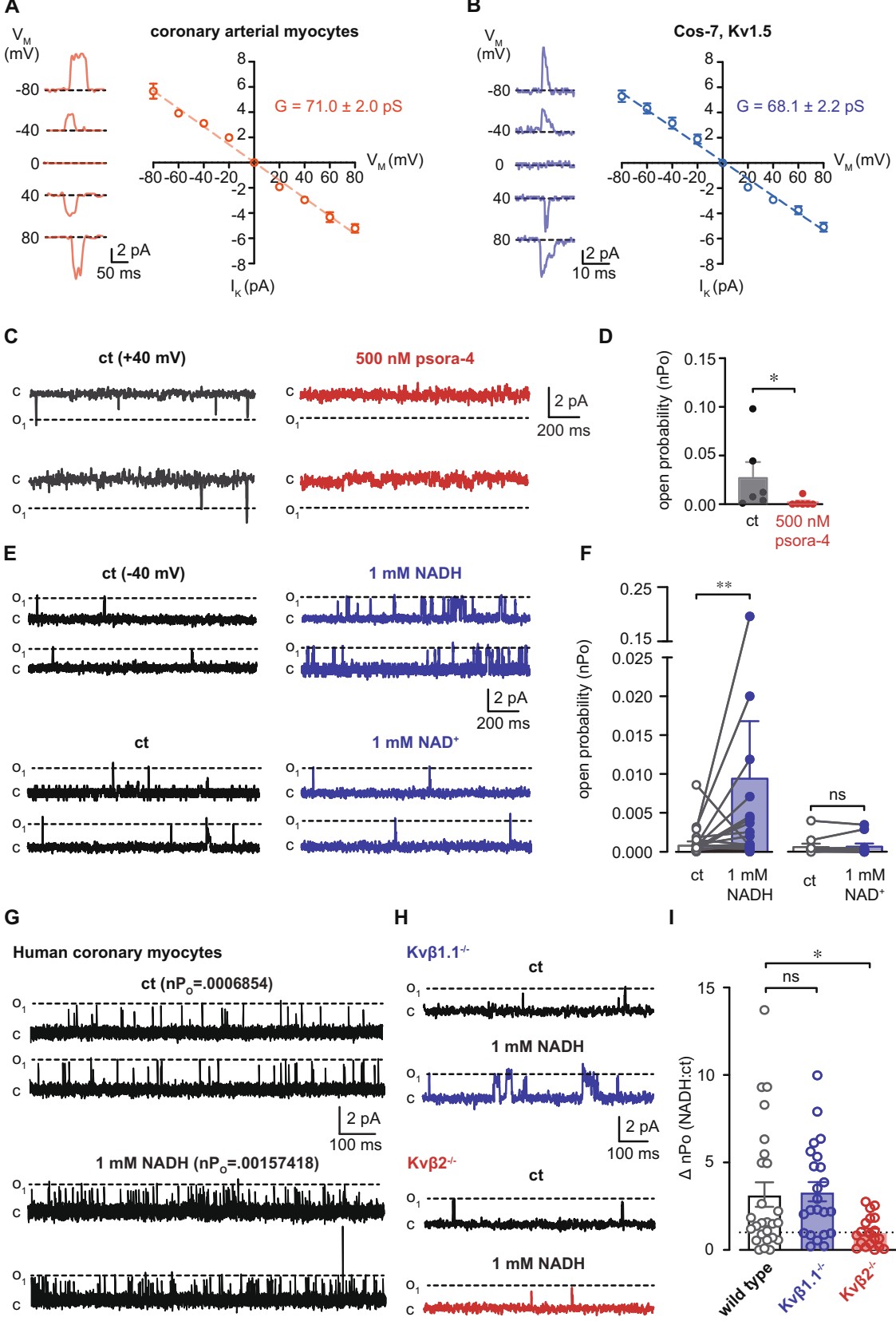

**Fig. 4 Potentiation of native coronary Kv1 activity by NADH requires Kvβ2. A**, **B** Unitary K$^+$ channel currents (-80-80 mV as indicated, *left*) and summarized I-V relationships (*right*) from inside-out patch recordings from isolated coronary arterial myocytes (wild type 129SvEv, **A**) or Cos-7 cells transiently expressing Kv1.5 (**B**). (**A**) $n = 6$ cells; (**B**) $n = 4$ cells. **C** Single K$^+$ channel activity in patches from coronary arterial myocytes at a holding potential of +40 mV in the absence and presence of psora-4 (500 nM). **D** Summary of K$^+$ channel open probabilities (nPo, mean values ± SEM) recorded in the absence and presence of 500 nM psora-4. $n = 6$ cells, *$p = 0.0312$ (two-sided Wilcoxon matched-pairs signed rank test). **E** Representative single Kv current recordings (holding potential = −40 mV) in the absence (ct) and presence of 1 mM NADH (*top*) or 1 mM NAD$^+$ (*bottom*). **F** Summary of Kv channel nPo (mean values ± SEM) recorded before (ct) and after bath application of 1 mM NADH ($n = 27$ cells) or NAD$^+$ ($n = 15$ cells). **$p = 0.0076$ (paired two-sided *t* test). **G** Single Kv channel currents recorded from freshly isolated human coronary arterial myocyte membrane patches before and after application of 1 mM NADH. nPo values for each condition are provided above each set of example traces. Data are representative of four independent experiments (one donor). **H** Inside-out patch recordings in coronary arterial myocyte membrane patches from Kvβ1.1$^{-/-}$ or Kvβ2$^{-/-}$ mice before and after application of 1 mM NADH. **I** Summary of fold-change in nPo (NADH:ct; mean values ± SEM) in patches from wild type ($n = 26$ cells from 8 mice), Kvβ1.1$^{-/-}$ ($n = 23$ cells from 8 mice), and Kvβ2$^{-/-}$ mice ($n = 19$ cells from 6 mice). *$p = 0.0115$ (Brown-Forsythe ANOVA with Dunnett's multiple comparisons test of log-transformed data). Source data are provided as a Source Data file.

workloads, as described previously[9,10]. Intravenous infusion of norepinephrine led to similar increases in mean arterial pressure and heart rate in both wild-type and Kvβ2$^{Y90F}$ mice (Fig. 5F). In wild-type mice, increases in cardiac workload (double product of mean arterial pressure × heart rate) were accompanied by a proportional increase in myocardial blood flow (Fig. 5G). Nonetheless, myocardial blood flow across the range of observed cardiac workloads was significantly suppressed in Kvβ2$^{Y90F}$ mice. The resultant suppression of blood flow upon loss of Kvβ2 catalytic function was comparable to that observed in mice lacking Kv1α subunits (i.e., Kv1.5 or Kv1.3) or Kvβ2. Collectively, these data indicate that pyridine nucleotide binding and catalytic cycling of Kvβ2 mediates the upregulated Kv1 activity underlying physiologic vasodilation and enhanced perfusion of the heart during acute metabolic stress.

## Discussion

The functional upregulation of voltage-gated K$^+$ channels in the coronary vasculature mediates metabolic hyperemia in the heart, yet the molecular events underlying how these channels sense changes in myocardial metabolism to dynamically regulate coronary arterial tone are unknown. In this study, we show that cardiac workload-dependent changes in the redox potential of pyridine nucleotides in coronary arterial myocytes enhance Kv1 activity and promote vasodilation via the auxiliary Kvβ subunit complex (Fig. 6). Catecholamine-driven increases in cardiac work in vivo and high frequency stimulation of cardiomyocytes ex vivo increase the cytosolic NADH:NAD$^+$ ratio in proximal arterial myocytes, consistent with this redox couple serving as a modifiable vascular effector of myocardial oxygen demand. Elevated levels of reduced pyridine nucleotides in coronary arterial myocytes promoted increases in Kv1 current density; and reduced (i.e., NADH), but not oxidized (i.e., NAD$^+$) nucleotides directly stimulated the activity of single native coronary Kv1 channels. Moreover, these effects were dependent on the presence of Kvβ2. Consistent with the concept that changes in pyridine nucleotide redox state and Kvβ2-mediated cofactor oxidation are required for nucleotide-sensitive upregulation of Kv1 activity, the ablation of Kvβ2 catalytic function abolished NADH-induced potentiation of Kv1 activity and vasodilation and suppressed myocardial blood flow.

The mechanical workload of the heart and rates of myocardial oxygen consumption are relatively stable at rest, yet can increase dramatically in response to environmental or physiologic cues. Resting heart rates in a majority of humans normally are between 50–82 beats per min[30], although periods of extreme tachycardia can be sustained with rates up to 240 beats per min without lasting structural damage[31]. Likewise, conscious mice have heart rates of 500–700 beats per min and can be increased to as high as 840 beats per min[32]. Such large ranges of myocardial work among

mammals are enabled by the intrinsic flexibility of myocardial metabolic activity. Under stress, myocardial oxygen consumption can increase as much as 10-fold relative to resting rates[33]. Nonetheless, sustaining high metabolic rates during stress requires near instantaneous and sustained increases in oxygen supply via arterial blood flow. Although extensive work has aimed to identify molecular processes regulating coronary perfusion as a function of cardiac metabolism, the underlying mechanism remains a fundamental enigma in cardiovascular physiology[33,34].

Resistance-sized arteries and arterioles of the coronary circulation maintain a partially constricted state from which they can readily dilate in response to local metabolic signals. In the absence of microcirculatory dysfunction, myocardial perfusion can increase 4–5-fold via reduced vascular resistance under conditions of high metabolic demand such as strenuous physical activity. To match oxygen supply to consumption, the coronary resistance arteries and arterioles modify their degree of tone in an orchestrated manner by independently responding to metabolic, hemodynamic, and neurohumoral signals[2]. In particular, the sensing of changes in local tissue metabolism by arterioles directly controls the perfusion of the low-resistance coronary capillary bed[35]. Previous work has shown that small arterioles regulate their diameter in response to changes in local shear stress (flow-mediated dilation), which depends upon endothelium-derived nitric oxide, particularly in smaller arteries[36,37]. While current consensus is lacking, it is thought that coronary vascular resistance is dynamically regulated by the integration of local physical forces (pressure and flow), neurohumoral modulation, and exposure to vasodilator metabolites such as adenosine, $pO_2$, and pH. How these factors influence the function of key effector pathways known to modulate coronary arterial myocyte function and vascular tone such as voltage-gated K$^+$ channels remains unknown. The findings presented here provide evidence that the capacity of vascular Kvβ proteins to sense pyridine nucleotide redox state and differentially control Kv1 activity is a missing link in the processes that couple myocardial metabolism with coronary arterial diameter and blood flow.

Previous work has shown that pharmacological inhibitors of Kv1 channels (4-aminopyridine or correolide) or genetic deletion of pore proteins that assemble Kv1 channels (Kv1.5 or Kv1.3) dissociate changes in blood flow from metabolism, resulting in ischemia and cardiac pump dysfunction[8,9,26]. Therefore, it seems that coronary blood flow is regulated primarily via Kv1 channels, which could be heteromeric assemblies of distinct Kv1 proteins[10,27,38]. Presumably, the formation of such heteromeric assemblies imparts greater functional diversity to Kv1 channels. This diversity of function could be further enhanced by binding to Kvβ proteins with divergent properties within the same channel complex. We have reported previously that Kvβ2 promotes, while Kvβ1 inhibits oxygen-dependent vasodilation[10];

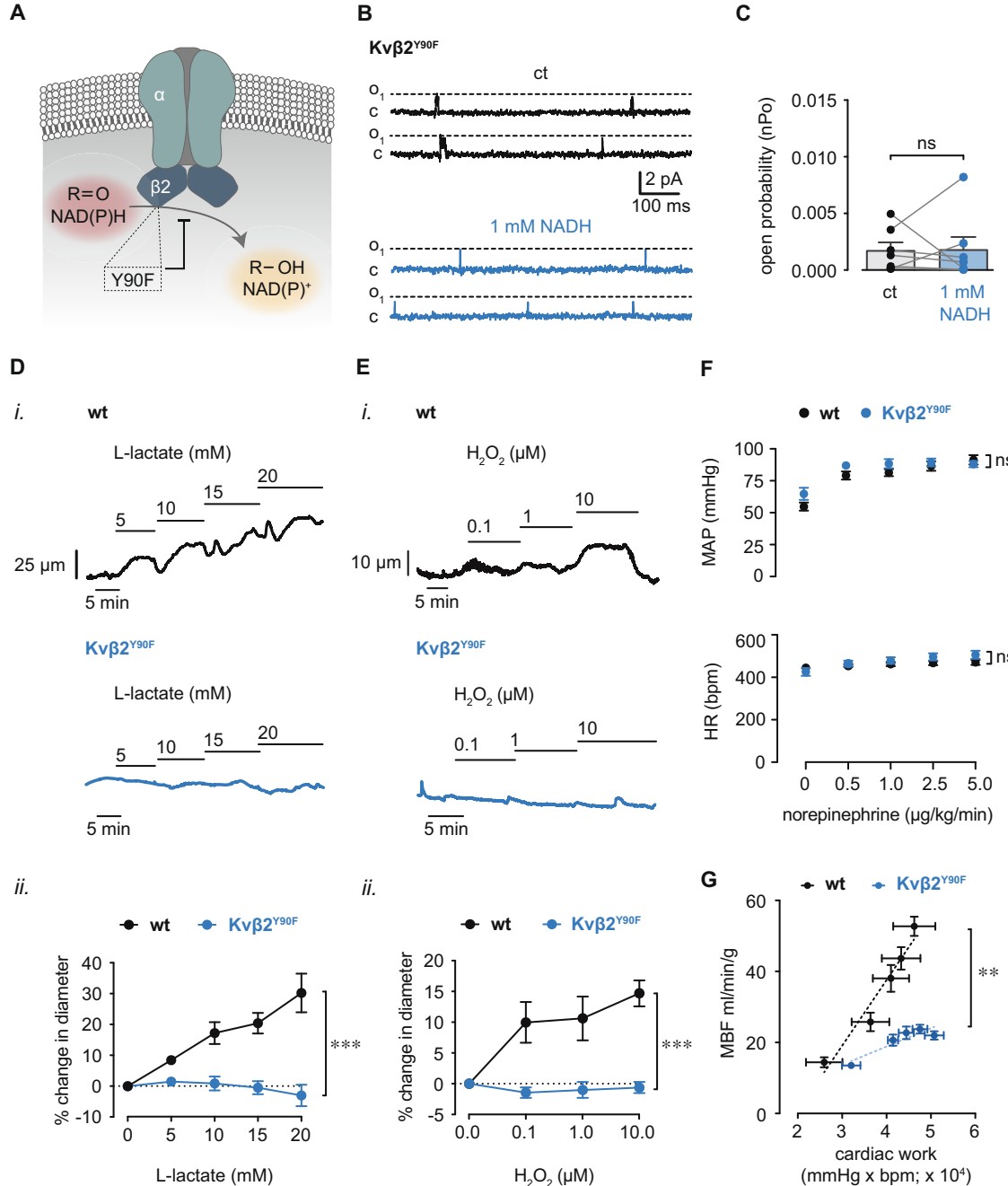

**Fig. 5 Loss of Kvβ2 catalytic function prevents redox-mediated increases in Kv1 activity and vasodilation, and suppresses MBF. A** Schematic illustration depicting abolished NADPH-dependent carbonyl reduction by Kvβ2 resulting from the Y90F point mutation. **B** Single-channel recordings showing Kv1 activity in membrane patches from Kvβ2$^{Y90F}$ mice before and after application of 1 mM NADH. **C** Summary of Kv1 open probabilities (nPo; mean values ± SEM) in membrane patches from Kvβ2$^{Y90F}$ mice before and after application of 1 mM NADH. $n = 7$ cells from 3 mice; ns: $P \geq 0.05$ (two-sided Wilcoxon matched-pairs signed rank test). **D** Representative arterial diameter recordings in arteries from wild type (129SvEv) and Kvβ2$^{Y90F}$ mice in the absence and presence of L-lactate (5–20 mM) in the perfusate (i.); and, summary of percent change in diameter (mean values ± SEM) in response to L-lactate (5–20 mM) relative to baseline (– L-lactate, ii.). wt: $n = 5$ arteries from 4 mice, Kvβ2$^{Y90F}$: $n = 6$ arteries from 5 mice. ***$p = 0.00001$ (two-way repeated measures ANOVA). **E** Representative arterial diameter recordings in arteries from wild type (129SvEv) and Kvβ2$^{Y90F}$ mice in the absence and presence of H$_2$O$_2$ (0.1–10 μM) in the perfusate (i.); and, summary of percent change in diameter (mean values ± SEM) in response to H$_2$O$_2$ (0.1–10 μM) relative to baseline (– H$_2$O$_2$, ii.). wt: $n = 4$ arteries from 4 mice, Kvβ2$^{Y90F}$: $n = 4$ arteries from 3 mice. ***$p = 0.0004$ (two-way repeated measures ANOVA). **F** Summary (mean values ± SEM) of mean arterial pressure (MAP) and heart rate (HR) in wild type and Kvβ2$^{Y90F}$ mice during intravenous norepinephrine infusion (0–5 μg/kg/min). $n = 9$-15 mice, ns: $P \geq 0.05$ (two-way repeated measures ANOVA). **G** Summary (mean values ± SEM) of relationships between MBF (ml/min/g) cardiac work (pressure rate product; bpm × mmHg) in wild type (wt; 129SvEv) versus Kvβ2$^{Y90F}$mice. **$p = 0.0043$, slope wt vs. Kvβ2$^{Y90F}$, $n = 4$-6 mice, (linear regression). Source data are provided as a Source Data file.

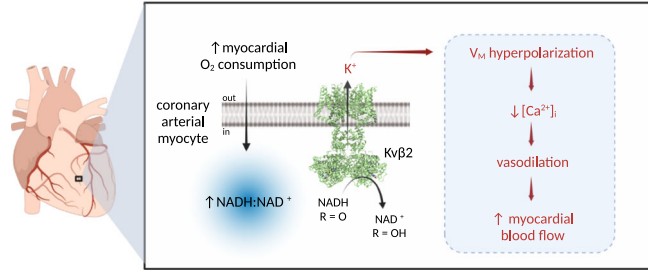

**Fig. 6 Oxygen demand-sensitive coronary arterial myocyte NADH:NAD⁺ links blood flow to myocardial metabolism via Kvβ2 catalytic function.** Schematic representation showing proposed model of myocardial blood flow regulation via coronary pyridine nucleotide redox potential. Increases in myocardial oxygen consumption and cardiac workload promote the rapid elevation of intracellular NADH:NAD⁺ in arterial myocytes in the wall of intramyocardial coronary arteries. This change in redox balance modulates the voltage-sensitivity of Kv1 gating via binding and enzymatic oxidation of NADH by Kv-associated Kvβ2 proteins. Resultant enhancement of steady-state K⁺ efflux leads to membrane potential (V$_M$) hyperpolarization, reduction in intracellular [Ca$^{2+}$]$_i$, arterial myocyte relaxation, and vasodilation, thus increasing blood flow. Created with BioRender.com.

therefore Kv1-Kvβ2 complexes of differing subunit compositions could generate a wide range of K⁺ currents, fine-tuned to the local demands and the metabolic state of the tissue. Nevertheless, the mechanism by which Kvβ2 senses metabolic changes and confers oxygen-sensitivity to Kv1 channels remains unclear. The Kvβ subunits bind to pyridine nucleotides with high affinity[16], and they catalyze hydride transfer to carbonyl substrates at low rates[29]. Because the binding of pyridine nucleotides to Kvβ alters Kv gating[28], changes in their concentration in response to changes in pO$_2$ could in principle regulate Kv1 activity. Indeed, we found that elevated tissue demand for oxygen increases the NADH:NAD⁺ ratio in arterial myocytes and that the ratios of pyridine nucleotides achieved at low pO$_2$ augment Kv1 activity.

Most cells maintain a low NADH:NAD⁺ ratio under aerobic conditions. However, even marginal decreases in O$_2$ availability reduce the rate of oxidative phosphorylation and can lead to the accumulation of NAD in its reduced form—i.e., NADH. This change in the redox state of the NADH/NAD⁺ couple is closely reflected in the lactate:pyruvate ratio, which is determined in part by the activity of lactate dehydrogenase operating near equilibrium conditions[19,39]. Aside from changes in O$_2$ availability, altered NADH:NAD⁺ induced by diffusible metabolites such as ROS or extracellular lactate results in vasodilation and this functional response is critically dependent on the expression and catalytic function of Kvβ2 (Fig. 5 and [10]). Loss of lactate-induced vasodilation in arteries from Kvβ-null animals indicates that the Kvβ complex itself, rather than the Kv pore domain, imparts pyridine nucleotide redox sensitivity to native vascular Kv1 channels. Moreover, our observation that the activity of Kv channels in coronary arteries is regulated directly by NADH:NAD⁺ in proportion to ratios expected under low and high oxygen tensions, and that regulation is abolished by the loss of Kvβ2, provide compelling evidence of the unique redox sensitivity of the Kv1-Kvβ2 complex and its specific dependence on intracellular pyridine nucleotide redox state.

It is notable that the effects of reduced pyridine nucleotides (i.e., NADH) on Kv1 activity extend to the coronary circulation of humans. Indeed, like murine coronary arterial myocytes, human coronary arteries express Kvβ proteins, which associate with Kvα pore proteins (see Supplementary Fig. 7 and ref. [40]). In large animals and small rodent models, Kv channels have been identified to be key effectors mediating the relationship between myocardial oxygen consumption and perfusion; however, other

redox sensitive cation channels (e.g., K$_{ATP}$, BK$_{Ca}$)[41] may be influenced by altered pyridine nucleotide levels to impact either basal vascular tone, or vasoactive responses to metabolic stimuli. Moreover, it is also unclear whether the mechanism reported here extends to vascular beds outside of the heart. Our previous work indicates that Kvβ proteins are expressed and functional in small arteries and arterioles of peripheral tissues (e.g., mesenteric vessels)[10]. Nonetheless, we speculate that organ-specific differences in tissue metabolism (e.g., basal and stress-related O$_2$ consumption) and associated metabolite generation may differentially impact smooth muscle pyridine nucleotide redox state, which our current study suggests is a key determinant of Kvβ function in the coronary vasculature. Additionally, the compositions of heteromeric Kvα:β complexes may be unique to a particular vascular bed to enable distinct functional responsiveness of Kv channels to changes in the local tissue environment.

Our observations reported here are consistent with prior findings suggesting a role for ROS signaling in regulating Kv activity in the coronary vasculature[6,42,43]. Elevated myocardial oxygen consumption has been found to be associated with increases in the levels of tissue hydrogen peroxide, a potent vasodilator that affects smooth muscle K⁺ channels[6]. Our data showing that the antioxidant tempol attenuates frequency-dependent elevation of NADH:NAD⁺ in arterial myocytes is consistent with a key role of ROS in regulating pacing-induced changes in pyridine nucleotide levels. Increases in ROS can acutely modify pyridine nucleotide redox ratios[44–46], which can then in turn regulate Kv activity. Moreover, our observation that vasodilation in response to hydrogen peroxide is abolished in arteries from Kvβ2$^{Y90F}$ mice provide strong evidence for a critical role of Kvβ. Nonetheless, additional work is required, not only to understand the relationship between hydrogen peroxide, pyridine nucleotide levels, and Kvβ-mediated catalysis, but also to determine whether increased ROS generation under conditions of high metabolic activity could be due to the activation of NAD(P)H oxidase and whether the effects of ROS are due to oxidative modification of Kvα or β proteins. However, a direct role of oxidative modification in regulating the function of these proteins seems unlikely in view of our observation that the effects of both hydrogen peroxide and lactate were abolished in arteries from Kvβ2$^{Y90F}$ mice, which suggests that NADPH-dependent catalysis per se, rather than oxidative modification of channel proteins is required for oxygen sensing.

Based on current findings, we propose that changes in the levels of reduced nucleotides upon increased oxygen consumption directly stimulates Kv activity, which is rapidly reversed upon catalytic cycling when nucleotide redox state returns to baseline. This mechanism is akin to that recently proposed in the excitability of Drosophila dorsal fan-shaped body neurons, in which ROS accumulation and the conversion of NADPH to NADP⁺ by Hyperkinetic (orthologue of mammalian Kvβ), underlies slow A-type current inactivation, action potential frequency enhancement, and the induction of sleep[47]. That a similar mechanism may be operative with respect to blood flow regulation in vivo is supported by evidence showing that acute increases in cardiac workload promotes the accumulation of the lactate:pyruvate ratio in the intramyocardial vasculature. Although we could not directly measure NADH:NAD⁺ ratio in vivo, the use of lactate:pyruvate ratio as a surrogate of pyridine nucleotide redox state is based on the tight coupled relationship between lactate:pyruvate and NADH:NAD⁺ ratios amply demonstrated in previous studies[19,39]. Thus, it is plausible that under conditions of increased workload, increased oxygen consumption and demand alters coronary arterial myocyte redox state in favor of reduced NADH:NAD⁺.

Despite strong evidence supporting this model, the specific processes underlying the workload-dependent effects on Kvβ

function remain unclear. Perhaps the most intriguing finding of our study is the dependence of nucleotide sensing on the catalytic activity of Kvβ. In other aldo-keto reductases, the apo-protein can bind to pyridine nucleotides in the absence of substrate and the nucleotide is released only after its oxidation; the release of the nucleotide is usually the rate-limiting step in the catalytic cycle[48]. A similar kinetic mechanism may be operating in Kvβ[29], which binds to nucleotides in the absence of any substrate[16]. However, for it to impact Kv gating, the cofactor must remain bound to the protein or undergo oxidation. Our observation that the regulation of Kv by Kvβ2 is abolished when the protein is catalytically incompetent favors the latter possibility—i.e., catalytic turnover of the protein is essential for the continued regulation of Kv gating under native conditions. Finally, even though this study did not aim to identify the in vivo substrates of Kvβ, our previous work suggests that lipid-derived aldehydes may be one class of potential candidates for such regulation[29,49]. Further investigations are required to identify the substrate(s) that support Kvβ catalysis and thereby regulate Kv activity and vasodilation under conditions of varied myocardial oxygen demand. Nevertheless, our current observations offer an example of a fundamental physiologic role for metabolic regulation of a vascular ion channel by direct catalytic activity of its subsidiary subunit. Our findings therefore constitute a unique paradigm for cellular oxygen sensing that may be an essential component of cardiovascular function in health and disease.

## Methods

**Animals and animal euthanasia**. All animal procedures were performed as approved by the Institutional Animal Care and Use Committees at the University of Louisville and Northeast Ohio Medical University. Genetically engineered mouse strains in which Kcnab1 or Kcnab2 genes were ablated (i.e., Kvβ1.1[−/−], Kvβ2[−/−], respectively)[50,51] or in which Tyrosine-90 of Kvβ2 was mutated to Phenylalanine (Kvβ2[Y90F])[51] were used for this study. Strain-matched wild-type mice (WT; C57BL6N for Kvβ1.1[−/−], 129SvEv for Kvβ2[−/−]) were used as controls (indicated in figure legends). Due to the potential for confounding effects of estrogen on the functional expression of vascular K+ channels[52–54], only male mice (aged-3–6 months) were used. All mice were bred and maintained in-house and fed normal chow ad libitum in a temperature-controlled room on a continuous 12:12 h light:dark cycle. Mice were euthanized by sodium pentobarbital (150 mg/kg; i.p.) and thoracotomy, and tissues were excised immediately for ex vivo assessments as described below.

**Human tissue**. Deidentified human tissues, obtained through the Midwest Transplant Network, were kindly provided for this study by Dr. Tamer Mohamed (Division of Cardiovascular Medicine, University of Louisville). Authorization was obtained according to the Uniform Anatomical Gift Act and IRB approval was not required because the project does not meet the definition of human subjects research. Branches of superficial and subepicardial left anterior descending coronary arteries were dissected from an excised heart (male, 52 years old; see Supplementary Table 1) in ice-cold Ca$^{2+}$-free physiological saline solution consisting of (in mM): 140 NaCl, 5 KCl, 2 MgCl$_2$, 10 HEPES, 10 glucose, pH 7.4. Arteries were enzymatically digested to isolate individual arterial myocytes as described below.

**Isolation of coronary arterial smooth muscle cells**. Hearts were excised and immediately transferred to ice-cold physiological saline solution containing (in mM): 134 NaCl, 6 KCl, 1 MgCl$_2$, 2 CaCl$_2$, 10 HEPES, and 7 glucose, pH 7.4. First and second-order branches of the left anterior descending coronary artery were manually dissected and smooth muscle cells were isolated as previously described[27]. Briefly, arterial segments were incubated at 37 °C in Ca$^{2+}$-free physiological saline (composition described above), containing papain (1 mg/ml) and dithiothreitol (1 mg/ml) for 5 min with gentle agitation. Then, the solution was replaced with buffer containing trypsin inhibitor (1 mg/ml) and collagenase (type H, 1 mg/ml) and incubated at 37 °C for an additional 5 min with gentle agitation. For human tissues, arterial myocytes were digested in collagenase (type H, 3 mg/mL; Sigma Aldrich), elastase (1 mg/mL; Sigma Aldrich), and bovine serum albumin (10 mg/mL; Sigma Aldrich) at 37 °C for 30 min with gentle agitation. Digested tissues were washed three times with ice-cold enzyme-free buffer. Cells were liberated by gentle trituration with a flame-polished glass pipette and kept on ice until use.

**Imaging mass spectrometry**. Hearts from anesthetized (sodium pentobarbital, 50 mg/kg, i.p.) mice that were acutely treated with dobutamine (10 mg/kg, i.p.) or vehicle (phosphate-buffered saline; see Fig. 1) were rapidly frozen with liquid N$_2$ and immediately excised and stored at −80 °C. Thin sections (12 µm) of fresh frozen mouse hearts were obtained from a cryostat (Leica CM 1900) and thaw-mounted onto gold-coated stainless steel MALDI target plates. Serial sections were obtained for H&E staining on standard glass microscope slides. Four serial sections were obtained for each heart and used for imaging mass spectrometry (IMS). An automated sprayer (TM Sprayer, HTX Technologies) was used to apply the MALDI matrix to the tissues. 9-Aminoacridine (9AA, hydrochloride salt, Sigma-Aldrich A38401) was prepared at 5 mg/ml in 90% methanol and sprayed at 0.12 ml/min, 85 °C, and 700 mm/min stage velocity. Eight passes were deposited at 3 mm track spacing, alternating horizontal and vertical positions between passes. Metabolite images were acquired on a 9.4 T FT-ICR mass spectrometer (Bruker Solarix, Bruker Daltonics) in negative ionization mode at 20 µm spatial resolution. Data were acquired in CASI mode, with Q1 isolating m/z 86 with a 6 amu window with a detected mass range of 72–350 amu. Images were visualized with Flex-Imaging software version 5.0.72.0_978_145 (Bruker). Regions of interest were selected to encompass the coronary wall and the surrounding tissue. Spectra were averaged for each region of interest, and the intensities for signals corresponding to lactate (m/z 89.0244) and pyruvate (m/z 87.0088) were exported.

**Smooth muscle and hiPSC-CM co-culture preparation and live-cell fluorescence imaging**. Vascular smooth muscle cells were isolated using previously described methods shown to yield highly pure CD31/CD45/lineage marker-negative and α-smooth muscle actin-positive cells[55]. Thoracic and abdominal aortae were excised and placed in ice-cold Dulbecco's Modified Eagle Media (DMEM) containing Fungizone™ (1:1000; Thermo Fisher Scientific). Vessels were cleaned of connective tissue, minced into 1 mm segments, and transferred to a sterile 15 ml tube and washed (3x) with cold Tyrode's solution containing (in mM): 126 NaCl, 44 KCl, 17 mM NaHCO$_3$, 1 MgCl$_2$, 10 Glucose, and 4 mM HEPES, pH 7.4. After washing, the solution was replaced with 1 ml of Tyrode's solution containing 1 mg/ml collagenase (Type 2, Worthington) and 20 µM CaCl$_2$. The tissue was incubated with intermittent agitation (4–5 h, 37 °C, 5% CO$_2$), centrifuged at 300g, and resuspended in DMEM containing 10% fetal bovine serum and 1% penicillin streptomycin. The cells were plated in 35 mM Primaria™ plates (Corning) for 5 days prior to use (P0-3) in imaging experiments.

Human-induced pluripotent stem cell-derived cardiomyocytes used for this study were purchased as differentiated myocytes (iCell Cardiomyocytes[2], FujiFilm Cellular Dynamics International; 01434; manufacturer's protocols) or derived from iPSCs in-house (iPSC line #SCVI15; Joseph Wu Laboratory, Stanford University). The SCVI15 iPSC line is previously established and deidentified; thus, the use of iPSCs in this study did not require IRB approval. Cryopreserved iPSCs were reprised and plated on Matrigel (Corning)-coated tissue culture grade dishes. The iPSCs were subsequently propagated in a feeder-free environment using StemFlex Medium (Thermo Fisher Scientific), replaced every two days, and maintained under standard incubation conditions (37 °C with 5% atmospheric CO$_2$). When approaching 85% confluency, iPSCs underwent clump cell passaging using Versene dissociation agent (Thermo Fisher Scientific). Cells were then differentiated using the iPSC Cardiomyocyte Differentiation Kit (Thermo Fisher) following the manufacturer's instructions. Briefly, at ~75–80% confluency, cells were stimulated to differentiate by the addition of pre-warmed Cardiomyocyte Differentiation Medium A (day 1). On differentiation day 3, medium was replaced with pre-warmed Cardiomyocyte Differentiation Medium B. At differentiation day 5, medium was replaced with pre-warmed Cardiomyocyte Maintenance Medium and replenished every other day until differentiation day 10. At this time, differentiated cells were subjected to metabolic selection using cardiomyocyte enrichment medium (48.1 mL of glucose-free RPMI 1640 Medium, 1.7 mL of Bovine Albumin Fraction V 7.5% w/v, 0.4 mL of 1 M sodium lactate, and 0.13 mL of 250x ascorbic acid solution). This solution was then replaced every other day for 5 days, after which, cultures were replenished with kit-supplied Cardiomyocyte Maintenance Medium with replacement every other day until use.

Plasmids for peredox-mCherry were obtained via Addgene (pcDNA3.1-Peredox-mCherry, #32383) and Ad-peredox-mCherry adenovirus was generated by insertion to an adenoviral backbone (Type 5, dE1/E3; Vector Biolabs). After reaching 50–80% confluency, arterial myocytes were treated with Ad-peredox-mCherry (6.4 × 10$^7$ PFUs, 50–100 MOI; 4 h at 37 °C). During infection and thereafter, cells were maintained in antibiotic-free DMEM supplemented with 2% FBS. For arterial myocyte-iPSC-CM co-cultures, Ad-peredox-mCherry-treated (24 h) arterial myocytes were seeded onto iPSC-CM monolayers, and the co-culture was maintained in DMEM supplemented with 5% FBS for 48 h prior to imaging.

Calibration of peredox-mCherry fluorescence for quantification of cytosolic NADH:NAD$^+$ was performed as previously described[21]. Briefly, cells were perfused with a baseline extracellular solution consisting of (in mM): 121.5 NaCl, 2 KCl, 25 NaHCO$_3$, 1.25 NaH$_2$PO$_2$, 1 MgCl$_2$, and 2 CaCl$_2$ (pH 7.4, maintained by aeration with 5% CO$_2$). Bath temperature was monitored in all experiments throughout with a thermistor probe and maintained at 36.5–37.5 °C. Green (ex 405, em 525) and red (ex 560, em 630) fluorescence was monitored during sequential application (10–15 min each) of lactate:pyruvate at ratios of 500, 160, 50, 20, and 6. At the end of each experiment, maximum and minimum green:red fluorescence was recorded in the presence of 10 mM lactate and 10 mM pyruvate, respectively. For experiments testing the effects of hypoxia and electrical stimulation of co-

cultures, cells were bathed in DMEM/5% FBS in an enclosed stage-top incubation system (Warner Harvard Apparatus) to allow equilibration of the bath solution to controlled O$_2$ levels (1–5%). Co-cultures and arterial myocytes alone (i.e., in the absence of iPSC-CMs) were electrically paced from 1–3 Hz using platinum electrodes connected to a MyoPacer field stimulator (IonOptix). Fluorescence images were acquired using a Keyence BZ-X800 epifluorescence imaging system with BZ-X800 Viewer software (version 01.01.01.03) in time-lapse mode with a 4X objective lens. Images were captured every 20 s in brightfield, green (T-sapphire), and red (mCherry) channels. Image series were converted to.avi files in Keyence Analyzer software (version 1.1.1.8) and analyzed using FIJI software (2.0.0-rc-69/1.53i; NIH).

**Patch Clamp Electrophysiology.** Coronary arterial myocytes were isolated from mice as described above. Outward K$^+$ currents were recorded using the conventional whole-cell configuration of the patch clamp technique in voltage clamp mode of an Axopatch 200B amplifier (Axon Instruments). Borosilicate glass pipettes were pulled using a P-87 micropipette puller (Sutter Instruments) to a resistance of 5–7 MΩ and filled with a pipette solution containing (in mM): 87 K$^+$-aspartate, 20 KCl, 1 MgCl$_2$, 5 Mg-ATP, 10 EGTA, 10 HEPES, pH 7.2. In some experiments, pyridine nucleotides were added into the internal solution (see Supplementary Table 2 for concentrations). Cells were allowed to adhere to a glass coverslip in a 0.25 ml recording chamber (Warner Instruments) and were bathed in a solution containing (in mM): 134 NaCl, 6 KCl, 1 MgCl$_2$, 0.1 CaCl$_2$, 10 Glucose, 10 HEPES. Series resistance was electronically compensated at ≥ 80%. Outward K$^+$ currents were recorded during a series of 500 msec step-wise depolarizations in 10 mV increments (−70 to +50 mV) from a holding potential of −70 mV. The voltage-dependence of activation was determined from tail currents elicited from repolarization to −40 mV. Voltage-dependence of inactivation was separately determined from a standard two-pulse voltage protocol in which cells were subjected to step-wise depolarizations (−100 to 50 mV) for 8 s followed by 200 ms pulse to 50 mV.

Single Kv channel activity was recorded using the inside-out configuration of the patch clamp technique with symmetrical bath/pipette K$^+$ conditions. Glass pipettes (8–10 MΩ) were filled with a solution containing (in mM): 140 KCl, 1 HEDTA, 10 HEPES, and 0.0001 iberiotoxin (pH 7.3 with KOH). The bath solution consisted of (in mM): 140 KCl, 1 HEDTA, 10 HEPES, and 0.001 glibenclamide. Excised membrane patches were held at a constant potential and stochastic channel activity was recorded in gap-free mode at a sampling frequency of 10 kHz. All electrophysiological data were analyzed using Clampfit software (version 10.6; Axon Instruments). Whole-cell current densities are expressed as the peak currents at each 500 msec depolarizing voltage step normalized to cell capacitance (pA/pF). $V_{0.5,act}$ and $V_{0.5,inact}$ are the voltages at half-maximum normalized current (I/I$_{max}$) determined from fitting data with a Boltzmann function. For single-channel analysis, open probabilities (nPo) were determined from recordings that had stable channel activity for at least 2 min. Values of nPo and amplitude were determined using the single-channel search function in Clampfit software (version 10.6; Axon Instruments).

**Echocardiography.** In vivo measurements of myocardial blood flow and cardiac function were performed as previously described[9,10]. Briefly, mice were anesthetized with isoflurane (3% induction, 1–2% maintenance; supplemental O$_2$ delivered at 1 L/min) and placed on a controlled heating platform in the supine position. A small incision was made on the right side of the neck for placement of a jugular venous catheter (sterile PE-50 tubing, prefilled with heparinized saline; 50 units/ml) to deliver contrast agent and drugs. For continuous monitoring of arterial blood pressure, a small incision was made on the hind limb and the femoral artery was isolated and cannulated with a 1.2 F pressure catheter (Transonic Systems), which was then advanced ~10 mm into the abdominal aorta. For myocardial contrast echocardiography (MCE), lipid-shelled microbubbles were prepared by sonication of decafluorobutane gas-saturated aqueous suspension of distearoylphosphatidylcholine (2 mg/mL) and polyoxyethylene-40-stearate (1 mg/mL), and intravenously infused at ~5 x 10$^5$ microbubbles/min. Imaging was performed using a Sequoia Acuson C512 imaging system (Siemens) with a high frequency linear array probe (15L8). A multi-pulse contrast specific pulse sequence was used to detect non-linear contrast signal at low mechanical index (MI = 0.18–0.25) and data were acquired during (i.e., destruction) and following (replenishment phase) a short 1.9 MI pulse sequence to destruct microbubbles within the acoustic field. Measurements were performed at baseline, after administration of the autonomic ganglionic blocker hexamethonium (5 mg/kg, i.v.), and subsequently after consecutive infusions of norepinephrine (0.5–5.0 μg/kg/min; 3 min each followed by 5 min washout). Mice that did not complete all norepinephrine infusions were excluded from the study. All data analyses and calculations of myocardial perfusion were conducted offline. Lab Chart 8 software (AD Instruments) was used for pressure and heart rate measurements.

For MCE, gain was adjusted to obtain images without myocardial signal in the absence of contrast agent infusion. Long axis images were acquired at a penetration depth of 2–2.5 cm. Images were analyzed to determine MBF by fitting intensity data from anterolateral regions of interest with an exponential function: $y = A(1-e^{-\beta t})$ where $y$ is the signal intensity at time $t$, $A$ is the signal intensity at plateau during the replenishment phase (reflects microvascular cross sectional volume) and

β is the initial slope corresponding to the volume exchange frequency[56]. Myocardial blood flow was estimated from 3 to 5 images per condition as the product of β x relative blood volume (RBV; myocardial to cavity signal intensity)[56,57]. MCE analyses were conducted in a genotype/treatment-blinded fashion.

**Ex vivo arterial diameter measurements.** Third and fourth-order branches of the mesenteric arteries were dissected and kept in ice-cold isolation buffer consisting of (in mM): 134 NaCl, 6 KCl, 1 MgCl$_2$, 2 CaCl$_2$, 10 HEPES, 7 D-glucose, pH adjusted to 7.4 with NaOH. Arteries were cleaned of connective tissue and used for diameter measurements within 8 h of isolation. Vessels were cannulated in cold isolation buffer on flame-polished glass micropipettes mounted in a linear alignment single vessel myograph chamber (Living Systems Instrumentation). After cannulation, the myography chamber was placed on an inverted microscope and arteries were equilibrated to temperature (37 °C) and static intralumenal pressure (80 mmHg), maintained with a pressure servo control unit (Living Systems Instrumentation) under continuous superfusion (3–5 mL/min) of physiological saline solution (PSS) consisting of (in mM): 119 NaCl, 4.7 KCl, 1.2 KH$_2$PO$_4$, 1.2 MgCl$_2$, 7 D-glucose, 24 NaHCO$_3$, and 2 CaCl$_2$ maintained at pH 7.35–7.45 via aeration with 5% CO$_2$, 20% O$_2$ (N$_2$ balanced). Following equilibrium (45–60 min), intralumenal diameter was continuously monitored and recorded with a charge coupled device (CCD) camera and edge detection software (IonOptix). Experiments were performed to examine the effects of L-lactate (Sigma Aldrich) (5–20 mM) and H$_2$O$_2$ (0.1–10 μM) in arteries preconstricted with the synthetic thromboxane A$_2$ analogue U46619 (100 nM; Tocris Bioscience). At the end of each experiment, the maximum passive diameter was measured in the presence of Ca$^{2+}$-free PSS containing the L-type Ca$^{2+}$ channel inhibitor nifedipine (1uM) and adenylyl cyclase activator forskolin (0.5 uM) as described previously[58,59]. Data were analyzed using IonWizard6.6 software and changes in diameter are expressed as the change from baseline (i.e., in the presence of 100 nM U46619) normalized to the difference between baseline and maximum passive diameters measured for each vessel.

**In situ proximity ligation.** Coronary arteries were enzymatically digested to isolate individual arterial myocytes, as described above, and allowed to adhere to glass microscope slides. After fixation with 4% paraformaldehyde (10 min, room temperature), cells were permeabilized with 0.1% triton-X100 and in situ proximity ligation was performed following the manufacturer's instructions. Briefly, non-specific antibody binding sites were blocked with the supplied blocking reagent and cells were labeled with primary antibodies against Kv1.5 (Neuromab, 75-011, 1:50), Kvβ1 (Abcam, Ab174508, 1:100), and Kvβ2 (Aviva Systems Biology, ARP37678-t100, 1:100) overnight at 4 °C. Cells were then treated with secondary oligonucleotide-conjugated probes, followed by ligation and rolling amplification by incubation with manufacturer-supplied ligase and polymerase, respectively. Sites of proximity were labeled with fluorophore (ex 554 nm, em 579 nm) and slides were mounted and sealed with coverslips. Brightfield and fluorescent images were captured using a 20x objective on a Keyence BZ-X800 All-in-One imaging system and fluorescent punctate sites and cell footprint area were quantified using FIJI software (2.0.0-rc-69/1.53i; NIH).

**Western blot.** Tissue lysates were obtained from mesenteric arteries (6–8 pooled segments of 3rd and 4th order branches) and homogenized in lysis buffer containing (in mM): 150 NaCl, 50 Tris-HCl, and 1 EDTA with 0.25% deoxycholic acid, 1% NP-40, and Complete Mini protease inhibitor cocktail (Roche; per manufacturer's instructions), pH 7.4. Tissue homogenates were sonicated on ice and centrifuged (10,000 x$g$, 10 min, 4 °C). Supernatants were transferred to another tube and boiled with Laemmli sample buffer (10 min) and run on a Stain-free Mini-PROTEAN 4–20% polyacrylamide gel (Bio-Rad). Total protein was assessed prior to transfer using a myECL imager (Thermo Fisher Scientific). Following transfer of proteins to polyvinylidene fluoride (PVDF), membranes were blocked for non-specific binding with 5% milk (wt/vol) in tris-buffered saline (TBS) and incubated (overnight, 4 °C) in primary antibodies against Kvβ2 (Neuromab, 75-021, 1:400) or α-tubulin (Sigma Aldrich, T5168, 1:4000) in TBS containing 0.1% Tween-20 (TBSt). After washing with TBSt (5x, room temperature), membranes were incubated with horseradish peroxidase (HRP)-conjugated secondary antibodies (anti-mouse IgG, Cell Signaling Technology, 7076, 1:3000). HRP was then detected with Pierce ECL Plus Western Blotting Substrate (Thermo Fisher Scientific) and a myECL imager (Thermo Fisher Scientific). Densitometry was performed for immunoreactive bands using FIJI software (version 2.0.0-rc-69/1.53i; NIH).

**Statistical analysis.** Group data are presented as mean ± SEM, unless otherwise indicated. Box-and-whiskers plots show the median (line), 25th to 75th percentile (box), and the min and max (whiskers). All data were analyzed with Prism 9 software (GraphPad Software) or SAS version 9.4 software (SAS Institute, Inc.). Normality was determined for datasets by Shapiro-Wilk tests. Specific tests used to compare experimental groups are provided in figure legends. For normal data, unpaired or paired two-tailed $t$ tests were used to compare two groups and one-way ANOVA with post-hoc tests, as indicated, were used for multiple comparisons of three or more groups. Linear mixed models were used to test for interactions in

time and genotype or treatment. For non-normal data, outcome variables were log-transformed for normality or nonparametric tests were used, as indicated. $P < 0.05$ was considered statistically significant.

**Study approval**. All animal procedures were conducted as approved by Institutional Animal Care and Use Committees at the University of Louisville and Northeast Ohio Medical University, in accordance with guidelines set by the National Institutes of Health.

**Reporting summary**. Further information on research design is available in the Nature Research Reporting Summary linked to this article.

## Data availability

All data supporting the findings described in this manuscript are available in the article and in the Supplementary Information, and from the corresponding author upon reasonable request. Source data are provided with this paper in a Source Data file. Source data are provided with this paper.

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

## Acknowledgements

This work was supported by grants from the National Institutes of Health (HL142710, GM127607, HL147921, M.N.), American Heart Association (16SDG27260070, M.N.), and the University of Louisville, School of Medicine (M.N.).

## Author contributions

M.M.D. designed experiments, collected, analyzed and organized data, prepared figures, and helped draft the manuscript, S.M.R., M.L.M., Z.B.W., G.M., and T.P. collected and analyzed data, D.W.R. performed and interpreted statistical analyses, M.L.R., V.O., and J.B.M. designed experiments, and collected and interpreted data, R.M.C. provided reagents and access to equipment and helped to revise the manuscript, B.H., W.M.C., and A.B. provided intellectual insight regarding experimental design and interpretation of data, and helped draft and revise the manuscript, M.A.N. conceived and directed the study, designed experiments, collected, analyzed and interpreted data, and drafted and revised the manuscript.

## Competing interests

The authors declare no competing interests.
