## [Peer Review File · Nature Communications]

REVIEWER COMMENTS

Reviewer #1 (Remarks to the Author):

This is an interesting piece of work with efficient IMS in physiology.

Reviewer #2 (Remarks to the Author):

The authors investigated the role of the Kvbeta2 subunit, and its regulation of Kv1 channels (especially Kv1.5) in regulation of coronary vasoreactivity and blood flow to the heart. Kvbeta2 contains a functioning aldo-keto reductase catalytic site and can therefore respond to the NADH:NAD⁺ ratio; increased pyridine nucleotide redox ratios indicative of increased oxygen demand were shown to augment Kv1.x potassium channel activity. This responsiveness was shown to require Kvbeta2, using Kvbeta2 null mice. The authors also used high-resolution MS imaging to quantify the NADH:NAD⁺ ratio in cardiac tissue. The authors have previously demonstrated a role for Kvbeta2 in regulating vasoreactivity and blood flow, but the present work extends this by demonstrating the role of changes in intracellular NADH/NAD⁺ ratio, and showing that this aspect was Kvbeta2-dependent. Although an extension of previous work in this area, the manuscript provides a substantive advance in our understanding of the ability of the cardiovascular system to respond to metabolic stress, especially that requiring increased heart rate and blood flow. The work is of high quality and I have only a few specific comments that need addressing.

1. Figure 4: panel C is mislabeled in the legend as panel D.

2. page 15 of the PDF, line 13: change to "the cytosolic".

3. The human heart rate range is quoted in the manuscript as 60-100 bpm. Athletes and those that exercise frequently very often have heart rates closer to 50 bpm.

4. How much of the Kvbeta2 effects are via Kv1.5 versus other channels? Other Kv1 channels or, e.g., Kv4 channels? Are there any data regarding null mice for those subunits, in this area of study? It would improve the Discussion to add some thoughts on this.

Reviewer #3 (Remarks to the Author):

This study by Dwenger et al. defines a potential new mechanism in the regulation of coronary artery function, which influences blood flow to the heart during metabolic stress. Specifically, they identify Kv β proteins, which are regulated by aldo-keto reductases and altered redox state of the pyridine nucleotide pool. This is an interesting and comprehensive study that has been well conducted.

However there are some concerns that warrant further consideration.

Specific comments

1. Human cardiac tissue was studied. Please provide some clinical information regarding the derivation of these samples. What was the clinical condition necessitating cardiac surgery?
2. The paradigm relating to changes in redox state that influence pyridine nucleotides secondary to metabolic stress is interesting. However a major deficiency in this study is the fact that there is no direct measurement of ROS levels or indices of oxidative stress. For example, H₂O₂ is known to induce vasorelaxation – how does this impact the paradigm presented?
3. This study focuses on K channels and cardiovascular functional changes in response to metabolic changes. However there are many ion channels sensitive to redox and metabolic stress important in coronary vasoreactivity. The potential role of other cation channels should be discussed and the rationale for focusing specifically on K channels should be delineated.
4. Considering the important role of NADP/NADPH in the generation of superoxide through activation of NAD(P)H oxidases (Noxs), it would be very useful to know how the paradigm presented

influences Nox-induced ROS production and redox state, which in turn influences NADP/NAD and K channel activity.

5. This study was conducted primarily in coronary vascular cells. Is this a cardiac-specific phenomenon or is this also evident in other vascular beds

Reviewer #4 (Remarks to the Author):

The authors have investigated the role of KV1.5 channel, and in particular its intracellular subunits Kv β 1.1 and Kv β 2 on coronary blood flow regulation. With a combination of in vivo and in vitro methods, they show that alterations in the NAD/NADH ratio modulate opening of the Kv1.5 channel via the Kv β 2 subunit, but only when it is catalytically active. Thereby they further delineate an important mechanism linking myocardial metabolism to coronary blood flow. The paper is convincingly written.

Major comments:

The authors describe that they have isolated coronary artery smooth muscle cells from one human patient. I may have missed it, but for which experiments were these used, as for the co-culture experiments, they use aortic smooth muscle cells, and for the patch-clamp studies murine coronary smooth muscle cells. Coronary flow is predominantly regulated in the coronary microcirculation. Is anything known about Kv β subunits in the (human) coronary microvasculature?

An increase in NADH/NAD ratio is also associated with an increase in oxidative stress, and may thereby induce oxidative modification of the channels. Please comment.

The data relating pacing frequency to NADH/NAD ratio (Figure 2D) appear to cluster in groups depending on the source of the cells, with two out of three subjects (triangles and squares) showing only a minor effect. I realize that these experiments are complicated, but although many cells were analyzed, with cells from only three subjects it is difficult to draw firm conclusions. Please increase n.

What is the definition of perivascular in figure 1c (which distance from the coronary blood vessels)? The authors show that the change in lactate occurs predominantly in the arterial wall, which to me

seems somewhat surprising, given that with the arterial myocytes being close to the blood stream, I expect that they will receive oxygen directly from the blood stream and hence equilibrate more to the arterial blood than to the surrounding tissue. Also, oxygen consumption of smooth muscle cells is relatively low, which would make it unlikely that their pO_2 decreases. Please comment.

Minor comments:

Please add scale bars to the histological pictures

Response to Reviewers

We thank the reviewers for their constructive feedback on our manuscript. Point-by-point responses to each of the reviewers' comments and concerns are provided below.

Reviewer #1:

This is an interesting piece of work with efficient IMS in physiology.

Thank you for the positive comment on our work.

Reviewer #2:

The authors investigated the role of the Kvbeta2 subunit, and its regulation of Kv1 channels (especially Kv1.5) in regulation of coronary vasoreactivity and blood flow to the heart. Kvbeta2 contains a functioning aldo-keto reductase catalytic site and can therefore respond to the NADH:NAD⁺ ratio; increased pyridine nucleotide redox ratios indicative of increased oxygen demand were shown to augment Kv1.x potassium channel activity. This responsiveness was shown to require Kvbeta2, using Kvbeta2 null mice. The authors also used high-resolution MS imaging to quantify the NADH:NAD⁺ ratio in cardiac tissue. The authors have previously demonstrated a role for Kvbeta2 in regulating vasoreactivity and blood flow, but the present work extends this by demonstrating the role of changes in intracellular NADH:NAD⁺ ratio, and showing that this aspect was Kvbeta2-dependent. Although an extension of previous work in this area, the manuscript provides a substantive advance in our understanding of the ability of the cardiovascular system to respond to metabolic stress, especially that requiring increased heart rate and blood flow. The work is of high quality, and I have only a few specific comments that need addressing.

1. Figure 4: panel C is mislabeled in the legend as panel D.

Thank you for pointing this out. This has been corrected.

2. Page 15 of the PDF, line 13: change to "the cytosolic".

Done.

3. The human heart rate range is quoted in the manuscript as 60-100 bpm. Athletes and those that exercise frequently very often have heart rates closer to 50 bpm.

Indeed, as the reviewer points out, resting heart rates in humans vary widely. We have revised this statement (see Discussion, 2nd paragraph) to reflect the *normal* range of resting heart rate. For instance, a recent study of 92,457 individuals wearing heart rate monitoring devices reported RHR of 65.5 ± 7.7 bpm, although the range among all individuals was 40 – 109 bpm.¹ In this study, 95% of men and women had resting heart rates in the range of 50 – 80 bpm and 53 – 82 bpm, respectively. Therefore, we have revised the statement in the Discussion to "*Resting heart rates in a majority of humans normally are between 50-82 beats per min, ...*" (see Discussion, paragraph 2).

4. How much of the Kvbeta2 effects are via Kv1.5 versus other channels? Other Kv1 channels or, e.g., Kv4 channels? Are there any data regarding null mice for those subunits, in this area of study? It would improve the Discussion to add some thoughts on this.

The Kv β proteins associate selectively with members of the Kv1 and Kv4 families. Transcripts encoding Kv4 subunits (i.e., Kv4.1-3) are detectable in various arterial beds (Xu et al., Am J Physiol., 1999, Nov; 277(5):G1055-63, Platoshyn et al., Am J Physiol., Lung Cell Mol Physiol., 2001, Apr; 280(4):L801-12);

however, whether Kv4 channels regulate vascular smooth muscle membrane potential, particularly in the coronary circulation, is not clear. Conversely, the role of Kv1 channels in the regulation of vascular smooth muscle membrane potential is well documented. In the coronary vasculature, genetic deletion of either Kv1.5 or Kv1.3 suppresses the relationship between myocardial workload and blood flow,^{4, 5} suggesting that Kv1 channels play a critical role in this response. This is in line with observations of reduced myocardial blood flow in animals treated with non-selective pharmacological inhibitors of Kv1.⁶ Thus, the effects of pyridine nucleotide sensing of Kv β 2 in coronary smooth muscle are likely mediated via Kv1 channels, which could be heteromers consisting of multiple α protein subtypes. We have revised the Discussion to clarify this point (see revised Discussion, paragraph 4).

Reviewer #3:

This study by Dwenger et al. defines a potential new mechanism in the regulation of coronary artery function, which influences blood flow to the heart during metabolic stress. Specifically, they identify Kv β proteins, which are regulated by aldo-keto reductases and altered redox state of the pyridine nucleotide pool. This is an interesting and comprehensive study that has been well conducted.

However, there are some concerns that warrant further consideration.

1. Human cardiac tissue was studied. Please provide some clinical information regarding the derivation of these samples. What was the clinical condition necessitating cardiac surgery?

As requested, available donor information associated with the necropsied human tissue used in our study, including medical history and cause of death, is now provided in **Supplemental Table 1** in the revised manuscript.

2. The paradigm relating to changes in redox state that influence pyridine nucleotides secondary to metabolic stress is interesting. However, a major deficiency in this study is the fact that there is no direct measurement of ROS levels or indices of oxidative stress. For example, H₂O₂ is known to induce vasorelaxation – how does this impact the paradigm presented?

Thank you for pointing this out. In our initial work, we focused on pacing-induced changes in pyridine coenzyme levels and how they affect Kv function. However, upon your suggestion, we probed further to see whether pacing-induced changes in pyridine-coenzymes could be due to changes in ROS generation. Indeed, as shown in Supplemental Figure S3, we found that treatment with the ROS scavenger TEMPOL blunted pacing-induced changes in pyridine nucleotides. These data suggest that pacing increases ROS production (either via increased mitochondrial activity or NAD(P)H oxidase activation), which may in turn affect Kv activity by modulating pyridine nucleotide levels. On the basis of these observations we speculated that hydrogen peroxide could also induce vasorelaxation by nucleotide redox sensing by Kv β 2. Consistent with this, we found that vasodilation in response to hydrogen peroxide was abolished in arteries from Kv β 2^{Y90F} mice. These results provide strong evidence for a critical role of Kv β in H₂O₂-mediated vasodilation (Figure 5E). Nevertheless, much work remains to be done, not only to understand fully the relationship between H₂O₂ and pyridine nucleotides in regulating vascular function, but also to determine whether covalent modification of channel cysteine residues could contribute to the effects of Kv β . However, the latter possibility seems remote because abolishment of the effect in Kv β 2^{Y90F} mice suggests that NADPH-dependent catalysis *per se* (rather than oxidative modification of the protein) is required for oxygen sensing.

The new results presented in Figures S3 and 5E and the Results and Discussion sections have been revised accordingly (see Results – pages 15 and 18; revised Discussion, paragraph 7).

3. This study focuses on K channels and cardiovascular functional changes in response to metabolic changes. However, there are many ion channels sensitive to redox and metabolic stress important in coronary vasoreactivity. The potential role of other cation channels should be discussed and the rationale for focusing specifically on K channels should be delineated.

We focused specifically on the role of Kv1 channels considering their well-documented role in the regulation of vascular smooth muscle membrane potential, redox sensitivity, and known central role in regulating metabolic hyperemia in the heart. The rationale for focusing specifically on Kv1 is delineated in our introduction. Nonetheless, we acknowledge the role of other vascular cation channels (e.g., K_{ATP} , BK_{Ca}) to the regulation of coronary vasoreactivity and the potential for their modulation by the reported pyridine nucleotide redox changes contributing to altered vascular tone and perfusion (see revised Discussion, paragraph 6).

4. Considering the important role of NADP/NADPH in the generation of superoxide through activation of NAD(P)H oxidases (Noxs), it would be very useful to know how the paradigm presented influences Nox-induced ROS production and redox state, which in turn influences NADP/NAD and K channel activity.

Our data do not exclude the possibility that changes in NAD(P)(H) during periods of elevated cardiac workloads may promote vascular-derived ROS via activation of NAD(P)H oxidases. Indeed, prior work has demonstrated that Nox2 is a significant source of ROS that contributes to the functional effects of endogenous vasodilatory agonists.^{7, 8} Further work is required to examine the hypothesis that changes in NAD(P)(H) secondary to metabolic stress, as reported in our current work, can modify the responsiveness of coronary arteries to vasoactive agonists via generation of Nox-derived ROS (see revised Discussion, paragraph 7).

5. This study was conducted primarily in coronary vascular cells. Is this a cardiac-specific phenomenon or is this also evident in other vascular beds?

Our previous work suggests that Kv β proteins are abundantly expressed in small arteries and arterioles of peripheral tissues (e.g., mesenteric vessels) and contribute to the functional regulation of Kv1 channels and vasoreactivity.³ Nonetheless, we speculate that organ-specific differences in tissue metabolism (e.g., basal and stress-related O₂ consumption) and associated metabolite generation may differentially impact smooth muscle pyridine nucleotide redox state, which our current study suggests is a key determinant of Kv β function in the coronary vasculature. Additionally, the compositions of heteromeric Kv α : β complexes are likely unique to the vascular bed in which they are expressed to enable distinct functional responsiveness to changes in smooth muscle metabolism. For instance, previous work shows that the pulmonary arterial tree expresses increasing levels of Kv β 1 protein progressively from larger conduit to smaller resistance vessels (Coppock et al., *Am J Physiol., Lung Cell Mol Physiol.*, 2001, Dec; 281:6): L1350-60), which may contribute to varying degrees of hypoxic pulmonary vasoconstriction between larger versus smaller vessels of the lung. We have expanded our discussion of this point in the revised manuscript (see revised Discussion, paragraph 6).

Reviewer #4:

The authors have investigated the role of KV1.5 channel, and in particular its intracellular subunits Kv β 1.1 and Kv β 2 on coronary blood flow regulation. With a combination of in vivo and in vitro methods, they show that alterations in the NAD/NADH ratio modulate opening of the Kv1.5 channel via the Kv β 2 subunit, but only when it is catalytically active. Thereby they further delineate an important mechanism linking myocardial metabolism to coronary blood flow. The paper is convincingly written.

1. The authors describe that they have isolated coronary artery smooth muscle cells from one human patient. I may have missed it, but for which experiments were these used, as for the co-

culture experiments, they use aortic smooth muscle cells, and for the patch-clamp studies murine coronary smooth muscle cells. Coronary flow is predominantly regulated in the coronary microcirculation. Is anything known about Kv β subunits in the (human) coronary microvasculature?

We apologize that this was not clear in our initial manuscript. Human coronary artery smooth muscle cells were used for proximity ligation and inside-out patch clamp experiments to examine Kv α/β interactions and the effects of NADH on single Kv1 channel activity, respectively (see **Figures S7 and 4G**). To our knowledge, only one previous study has examined the expression of Kv β subunits in small diameter human coronary arteries isolated from atrial tissue.² Using RT-PCR, the authors of this study provided evidence of Kv β 1.1, Kv β 1.2, and Kv β 1.3 transcript expression in human coronary arteries from patients with and without coronary artery disease. While this study did not probe for Kv β 2, our current study suggests that Kv β 2 proteins confer pyridine nucleotide redox sensitivity to native Kv1 channels in the setting of cardiac stress and metabolic hyperemia. This is now discussed in our revised manuscript (see revised discussion, paragraph 4).

2. An increase in NADH/NAD ratio is also associated with an increase in oxidative stress, and may thereby induce oxidative modification of the channels. Please comment.

This is an excellent point. Please see our response to comment #2 by Reviewer 3. As stated there - we found that treatment with the superoxide dismutase (SOD) mimetic TEMPOL blunted pacing-induced changes in pyridine nucleotides. These data suggest that pacing increases ROS production (either via increased mitochondrial activity or NAD(P)H oxidase activation), which in turn affect Kv activity by modulating pyridine nucleotide levels. On the basis of these observations we speculated that hydrogen peroxide could also induce vasorelaxation by affecting pyridine nucleotide levels. We tested this possibility, and found that vasodilation in response to hydrogen peroxide was abolished in arteries from Kv β 2^{Y90F} mice. These results provide strong evidence for a critical role of Kv β in H₂O₂-mediated vasodilation. Further work is required to determine whether an increase in oxidative stress may lead to covalent modification of the channel. Although we cannot rule out this possibility as Kv channels have reactive and accessible cysteine residues, our observation that vasodilation by hydrogen peroxide was abolished in Kv β 2^{Y90F} mice suggests that catalytic activity of the protein, rather than covalent modification, is required for oxygen sensing.

3. The data relating pacing frequency to NADH/NAD ratio (Figure 2D) appear to cluster in groups depending on the source of the cells, with two out of three subjects (triangles and squares) showing only a minor effect. I realize that these experiments are complicated, but although many cells were analyzed, with cells from only three subjects it is difficult to draw firm conclusions. Please increase n.

Based on this concern, we performed additional experiments to increase n. These new data strengthen our conclusions and are provided in Figure 2D.

4. What is the definition of perivascular in figure 1c (which distance from the coronary blood vessels)? The authors show that the change in lactate occurs predominantly in the arterial wall, which to me seems somewhat surprising, given that with the arterial myocytes being close to the blood stream, I expect that they will receive oxygen directly from the blood stream and hence equilibrate more to the arterial blood than to the surrounding tissue. Also, oxygen consumption of smooth muscle cells is relatively low, which would make it unlikely that their pO₂ decreases. Please comment.

We have revised the legend for Figure 1 to define the perivascular region of interest in MALDI-MS experiments. Note that our data indicate that high cardiac workloads elevate lactate:pyruvate ratios in both the arterial wall as well as the perivascular myocardium (see **Figure 1D**). While increased oxygen consumption by the myocardium accompanies elevated workload, we argue that changes in lactate:pyruvate and pyridine nucleotide redox state in the coronary wall can result without changes in local

pO₂; conversely, it is plausible that this change could result from generation of diffusible metabolites that are taken up by smooth muscle cells and influence metabolism independent of local pO₂ gradients. Indeed, this is supported by new data added to the revised manuscript suggesting that generation of ROS in oxygen-controlled cardiomyocyte-arterial myocyte co-cultures could, at least partially, contribute to changes in smooth muscle NADH:NAD⁺ upon increased pacing frequencies (see new **Figures 2D** and **S3**).

Minor comments:

1. Please add scale bars to the histological pictures

We apologize for this oversight. Scale bars have been added to images in the revised manuscript.

Literature cited

1. Quer G, Gouda P, Galarnyk M, Topol EJ, Steinhubl SR. Inter- and intraindividual variability in daily resting heart rate and its associations with age, sex, sleep, BMI, and time of year: Retrospective, longitudinal cohort study of 92,457 adults. *PLoS one* **15**, e0227709 (2020).
2. Nishijima Y, *et al.* Shaker-related voltage-gated K(+) channel expression and vasomotor function in human coronary resistance arteries. *Microcirculation* **25**, (2018).
3. Ohanyan V, *et al.* Myocardial Blood Flow Control by Oxygen Sensing Vascular Kvbeta Proteins. *Circulation research*, (2021).
4. Ohanyan V, *et al.* Requisite Role of Kv1.5 Channels in Coronary Metabolic Dilation. *Circulation research* **117**, 612-621 (2015).
5. Ohanyan V, *et al.* Kv1.3 channels facilitate the connection between metabolism and blood flow in the heart. *Microcirculation* **24**, (2017).
6. Goodwill AG, *et al.* Critical contribution of KV1 channels to the regulation of coronary blood flow. *Basic Res Cardiol* **111**, 56 (2016).
7. Larsen BT, Bubolz AH, Mendoza SA, Pritchard KA, Jr., Gutterman DD. Bradykinin-induced dilation of human coronary arterioles requires NADPH oxidase-derived reactive oxygen species. *Arteriosclerosis, thrombosis, and vascular biology* **29**, 739-745 (2009).
8. Zhou Z, *et al.* Involvement of NADPH oxidase in A2A adenosine receptor-mediated increase in coronary flow in isolated mouse hearts. *Purinergic Signal* **11**, 263-273 (2015).
9. Rettig J, *et al.* Inactivation properties of voltage-gated K⁺ channels altered by presence of beta-subunit. *Nature* **369**, 289-294 (1994).

REVIEWERS' COMMENTS

Reviewer #2 (Remarks to the Author):

The authors' revisions have satisfactorily addressed my previous comments.

Reviewer #3 (Remarks to the Author):

The authors have adequately addressed my previous concerns. The paper is significantly improved. This is an interesting study.

Reviewer #4 (Remarks to the Author):

Thank you for your careful revision of the manuscript. I have no further comments.